# Dinner Table Experience in the Flyover Provinces: A Bricolage of Rural Deaf and Disabled Artistry in Saskatchewan

Chelsea Temple Jones [1,*], Joanne Weber [2], Abneet Atwal [1] and Helen Pridmore [3]

1    Department of Child and Youth Studies, Brock University, St. Catharines, ON L2S 3A1, Canada
2    Faculty of Education, University of Alberta, Edmonton, AB T6G 2R3, Canada
3    Independent Researcher, Saskatoon, SK S7N 0G3, Canada
*    Correspondence: cjones@brocku.ca

**Abstract:** "Dinner table experience" describes the uniquely crip affect evoked by deaf and disabled people's childhood memories of sitting at the dinner table, witnessing conversations unfolding around them, but without them. Drawing on 11 prairie-based deaf and/or disabled artists' dinner table experiences, four researcher-artivist authors map a critical bricolage of prairie-based deaf and disabled art from the viewpoint of a metaphorical dinner table set up beneath the wide-skyed "flyover province" of Saskatchewan. Drawing on a non-linear, associative-thinking-based timespan that begins with Tracy Latimer's murder and includes a contemporary telethon, this article charts the settler colonial logics of normalcy and struggles over keeping up with urban counterparts that make prairie-based deaf and disability arts unique. In upholding an affirmative, becoming-to-know prairie-based crip art and cultural ethos using place-based orientations, the authors point to the political possibilities of artmaking and (re)worlding in the space and place of the overlooked.

**Keywords:** disability and deaf art; dinner table experience; creative analytic practice; affirmative ethics; Canadian prairie

## 1. Introduction

Commonly, deaf[1] and disabled people carry childhood memories of sitting at the dinner table witnessing conversations unfolding around them but without them. The dinner table, a symbol of family experience sharing in Western hearing worlds, represents Anglo-civility, normate comportment, and the shared experience—for some, not others—of keeping up (Bahan et al. 2008). Meek (2018) describes feelings of isolation and frustration that come from being excluded from this bonding time: "The *dinner table syndrome* is a metaphor for all of the conversations that are not completely accessible when deaf people are in situations with hearing groups" (p. 2).[2] Knowing that this metaphorical experience relies on normative orientations toward the dinner table, we begin this experimental writing with one of Ahmed's (2006) opening questions in *Queer Phenomenology*: "How do we begin to know or to feel where we are, or even where we are going, by lining ourselves up with the features of the grounds we inhabit, the sky that surrounds us or the imaginary lines that cut through maps?" (p. 6).

This question is crucial to our orientations toward each other and the metaphorical dinner table because it directs our attention to what Braidotti (2018) calls the "non-negotiable politics of location" (p. 210). Taking place as both literal (where we are now) and ephemeral (the trajectories of thought we espouse), we suggest that location is a leading feature of affirmative-ethics-based process ontology and the eventual re-ordering of the world that is central to the creative and political work of deaf and disability arts. Following Chandler (2018) and this special issue's attention to (re)worlding—an affective, artful, and imaginative project guided by multiple lines of thought—we consider both the conditions of possibility for the establishment of difference in the Canadian Prairies as well as new

worldly arrangements made possible through deaf and disabled people's creative and political work. Ahmed's question points upward at "the sky that surrounds us," and is a reminder that the sky is culturally significant in Saskatchewan and for the (re)worlding that happens here. To find the "imaginary lines" that border Saskatchewan on the colonizer's map, plant your finger right in the middle of Canada. It will land on a rectangular landmass between the provinces of Alberta and Manitoba cropping a 651,000-kilometer space that engulfs six Treaty territories—Treaty 2, Treaty 4, Treaty 5, Treaty 6, Treaty 8, and Treaty 10. Much of our work takes place in Treaty 4, a negotiation between the Cree and Saulteaux in 1874 that remains important for the 11 First Nations that live in the treaty's catchment area (Stonechild n.d.). There is little to obstruct the view of the sky on this flat land, with its straight, snaking highways and Saskatchewan license plates emblazoned with the 1997 contest-winning slogan, "The land of living skies" (Romuld 2020). The livingness of the sky is sometimes characterized by bird migrations and cloudy streaks of exhaust left behind by flyover planes. Residents cheekily call Saskatchewan a "flyover province" because it is couched between urban destinations on either coast of Canada, marking an overlooked-ness that parallels the uniquely crip dinner table experience affect. Our imagined metaphorical dinner table exists beneath a wide sky on this flat, colonized land with rich but often overlooked disability art and culture.

Through this experimental writing, which draws on prairie-based deaf and disabled artists' descriptions of dinner table experience, we actively construct methods "from the tools at hand" rather than from pre-existing guidelines—an exploratory mode of scholarship Kincheloe (2005) calls bricolage (p. 324). We read bricolage as intimate scholarship because it involves process ontologies that attend to unexplored embodied knowledge through action (Strom et al. 2018), including "coming-to-know" through dialogue such as interviews, jam sessions, and creative analytic practice—methods we describe below. The process ontology represented in the writing that follows is an act of collective intimate scholarship that aims to account for the overlooked and calls for affirmative ways of becoming and (re)worlding through radical relationality amid interconnected events that shape our contexts (Braidotti 2019). Leading with location, we position ourselves as bricoleurs mapping contemporary upswells in deaf and disabled art in the northern center of Turtle Island known as the Canadian Prairies. Below, we elaborate on "coming-to-know" or, *becoming-to-know*, through a bricolage of partial, fragmented knowledges (Haraway 1988), collective memory work, and narrative—all amid settler perspectives and the intricate, un-pin-downable performative nature of rurality and voice that shapes a prairie-based crip art movement embedded in the political possibilities and (re)worlding efforts of the overlooked (Cella 2017; Hamilton et al. 2016, p. 183).

## 2. Positioning

Our writing collaboration includes Jones (she/her), a hearing white settler who spent much of her childhood on the Canadian Prairies growing up alongside a disabled brother; Weber (she/her), who is deaf and the artistic director of Deaf Crows Collective based in Regina, Saskatchewan; Atwal (she/her), a hearing first-generation South Asian Canadian based in the Region of Peel in Ontario; and Pridmore (she/her), a hearing first-generation British Canadian settler who is a musician based in the Prairies. We have been involved in this arts-based research collaboration since a 2019 event in Regina called "Disability Artivism in the Flyover Provinces," which began slowly mapping contemporary upswells in deaf and disabled art in Saskatchewan.

Following Braidotti (2017, 2018, 2021), who asserts that a politics of location demands accountability and new ways of thinking about processes of becoming, we situate ourselves as *becoming-to-know*. Becoming-to-know refers to practice-based processes of knowledge creation that are deeply attentive to context (Hamilton and Pinnegar 2014, 2015; Jakubik 2011) and that are "thinking alongside" rather than centering identity (Braidotti 2018, p. 210). For us, coming from fields of critical disability studies (Jones), deaf education (Weber), disabled children's childhood studies and DisCrit (Atwal), and music (Pridmore),

this means tracking our transdisciplinary process ontology (Yardley 2020). To do this, we embrace non-linearity and associative thinking (what Braidotti, following Derrida, calls "zig-zag thinking" (2018, p. 210) as we ongoingly build our (inevitably partial) understanding of the disability and deaf art scene on the Canadian Prairies, beginning in Saskatchewan. Here, we meet at the intersections of disability, gender, race, place, and voice to dwell in the uniquely crip affect of not keeping up evoked by memories of being overlooked individually and culturally. Following Hardy,[3] we recast "dinner table syndrome" as "dinner table experience" (to usurp pathological connotations of "syndrome") and we bricolage three key markers of prairie-based disability culture to lend context to our research and to ground the (re)worlding work that is ongoing in this place: the 1993 killing of Tracy Latimer, the ongoing existence of a major telethon, and amidst this, the dinner table experience affect that impacts the culminating rise of deaf and disability arts and the (re)worlding work of this movement along the way.

### 3. Context: Research in "the Flyover" Provinces

Imagine the scene: a fowl supper, prairie gothic style. A plastic, spill-proof tablecloth covers a long fold-out table in a church basement in the town of Wilkie, Saskatchewan—a former railway station on the Canadian Pacific Railway line, today a town with a population of just over 1300 people. It is October 1993, a few days after the murder of a 12-year-old disabled girl, Tracy Latimer. Weber, a long-time resident of the community, remembers:

> It is a grey day. Farmers are continually thwarted by rain, unable to get their crops off in a timely manner. A few days ago, a farmer snapped and parked his daughter in his truck, left the ignition on in order to induce death by carbon monoxide poisoning. Despite the meager supports available to the family (who was struggling with the care for the daughter, unable to move independently, and a newborn baby), the daughter was sacrificed against the economic, social, and personal survival of the family.

Soon, this murder will make national and global headlines, escalating public debate and polarization over whose lives are worth living. In November 1994, before the case goes to court, the monthly Canadian news magazine *Maclean's* releases a full-page cover that puts readers in Robert Latimer's position by boldly asking, "what would you do?" (*Maclean's*). As the court deliberates in 1997, a *New York Times* pull quote tells readers "[a] euthanasia case becomes a cause célèbre in Canada" (DePalma 1997, p. A3). By 2009, Robert Latimer is on parole and BBC News World Service is interviewing him for a documentary broadcast that airs across the globe (*The Interview* 2009). This story will cast a long shadow on disability culture in Canada, though its rural contours will often be overlooked—flown over, in a way, as urban-based critics and scholars condemn the killing.

For now, we have gathered at this community table to reflect on, and operationalize, the setting. We wonder if "the Prairies" refers to the sky or the land or the people who live here, or to something else—perhaps a thicker, deeper affective economy rooted in a prairie aesthetic. Marler (2011) describes "the Prairies" as "a landscape perceived, paradoxically, as both edenic and utopic as well as dystopic—foreign, empty, abject" (p. 9). Or, put another way, "the Prairies" operate in a culture of crisis despite its vividly romantic and picturesque living skies (Kaye 2005). Aware that power and place are coproduced (Tuck and McKenzie 2015), we know that the temporal and spatial dimensions of this work leave their mark on our bodies, and it is our bodies that move us through our *becoming-to-know* methodology and methods.

In our findings below, we situate our research processes in a Western, prairie-entangled cultural logic whose dominant narratives of deafness and disability result in a gendered, ableist, audiocentric, and rural cultural mix. Despite the perception that "nothing happens" while we occupy this space and place, what emerges as grievable is determined as that which is "culturally living" in the minds, memory, and events perceived by others (Massey 2005; Pullman and Nichols 2015, p. 29). Yet, given that non-normative lives have been recognized as lives to be grieved, disability and deaf cultural production is always bound

up in the complex politics of grief—grief is a line etched into the dinner table (Ahmed 2015; Piepzna-Samarasinha 2022). In building a bricolage deaf and disability artistry on the Prairies, the lingering grief around Tracy Latimer's murder serves as a useful foundation. This event flags a particular apprehension of space and time that frames our research: what sets Latimer's murder apart from other disability rights propellors in Canada is the gendered nature of this violence and its subsequent excusability based on white settler logics willing to overlook the killing of a disabled girl.

### 3.1. "A Perfectly Nice Guy": Convergences of Race, Gender, and Disability

To understand the conditions of possibility for Tracy Latimer's murder, we consider the construction of prairie-based white femininity in the late nineteenth century that helped sustain concepts of racial and cultural difference by establishing the vulnerable white woman and girl. Euro-American settlement began on the prairies over a century earlier through farming, which was—and remains—a patriarchal activity (Bye 2005; Corman and Kubik 2016). However, women have long been "deeply intertwined" with farming communities writ large, even across racial divides (Fletcher 2016, p. 261). During early settler contact, Indigenous women assisted newcomers with their knowledge of medicine, thereby strengthening community bonds between themselves and white women (Carter 1997). Yet, this history tends to be overshadowed by gendered colonial stories (Faye 2006). As Carter (1997) chronicles in *Capturing Women: The Manipulation of Cultural Imagery in Canada's Prairie West*, dominant tropes of white prairie women as frail and delicate contributed to settlement and civility work; the pure, pious woman's free labor was useful in "[emphasizing] the frailty and delicacy of the white woman, as well as her dependence on males" (p. 8). To compound matters, white women's perceived frailty paired well with an imagined threat of Indigenous people, alienating Indigenous and white women from one another (Carter 1997, p. 6). This resounding imaginary of the fragile, white prairie woman contributes to Saskatchewan's "unique mythology" of settler aesthetic that both influences subjectivity production and supports Canadian nation building (Marler 2011, p. 1).

Tracy was not exempt from this settler aesthetic. A farmer's daughter, she was also bound up in patriarchal myths that bind male farmers to the land—part of the great "Canadian outdoors" (Laurendeau et al. 2021)—and an inherently colonial impulse to tame it. In a chapter called "The Tantalizing Possibility of Living on the Plains," Kaye (2005) describes the prairie farmer when he is positioned as neighbor:

> It would be easy to play our neighbour as the heavy—the big John Deere tractor loafing diesel fumes as he goes home for lunch, passing a petition the first year we were here to close down the village school because the cost of accommodating a child who used a wheelchair might raise taxes too high. A big landowner, a successful farmer, bent on passing to his children a farm homesteaded by ancestors a century ago . . . It would be easy to say that the greed . . . that would refuse ramps to a child in a wheelchair (that would suggest, in fact, that the child should have his legs amputated so he could be fitted with prosthesis and taught to climb stairs) is monstrous greed . . . but such a metaphoric reading is itself cramped and cruel, no better than scapegoating. My neighbour is a perfectly nice guy. (p. 30)

Kaye's (2005) description of this neighbour-farmer demonstrates how ableism layers into the highly gendered and colonial characteristics of this man, whose violent thoughts and actions are excused on the basis that he is a "perfectly nice guy" (p. 30). By Kaye's account, the "perfectly nice guy" is a character who embodies a cultural ethos where white male dominance settles both lands and disputes over personhood. To observe how this ethos translates into reality, we recall that in 2018 white male farmer Gerald Stanley was found not guilty of second-degree murder by a jury in Battleford, Saskatchewan after shooting Indigenous man Colten Boushie as he sat in the back of a parked vehicle. In a *New York Times* op-ed, Saskatchewan-born Indigenous writer Scrimshaw called out "the

century-long history of systemic racism that led to Colten Boushie's killing" (Scrimshaw 2018).

If conditions of possibility for Boushie's recent murder and the decided innocence of his killer have roots in over a hundred years of racism, then it is possible that the conditions of possibility for Tracy Latimer's killing also lay in the historic legacy of the "perfectly nice guy" insofar as this imagined subject is contingent on the intersection of disability, gender, race, and place. After all, Janz (2009) argues that early-aughts media coverage of Tracy's killing overemphasized unwarranted descriptions of pain conflated with cerebral palsy so thoroughly that Tracy's personhood was subsumed by ableist presumptions of suffering (p. 46).[4] Today, Robert Latimer's ambition of having his earlier conviction for second-degree murder overturned is supported by a public that describes him as a "simple" man up against "irrational" critics (Issa 2019; Janz 2018; Bauslaugh 2010). By contrast, Tracy was on a trajectory toward becoming a disabled white woman in a Western context, where gender and coloniality are significant forms of cultural capital (Thobani 2018; Fletcher et al. 2020) and where the work of social reproduction may be inaccessible for some disabled women. The risks of not contributing to capitalist production are high for disabled folks (Goodley et al. 2014). And, in Canada, a country that actively relies on the reproduction of inspirational disability tropes to uphold settler dominance (Peers 2015), Tracy's white, rural femininity/fragility also overlapped with tropes of innocence already etched into childhood (Dyer 2019). This case lays bare the collision of well-entrenched prairie tropes at the intersection of disability, gender, race, and place: the absence of Tracy's personhood combined with the apparent criminal immunity afforded to some white prairie men positions Robert Latimer as a "perfectly nice guy" intervening in the life, and ultimately the death, of a frail prairie girl.

However, prairie-based disabled artists have narrativized Tracy's case differently. Following deaf and disability arts traditions of offering multiple representations of existent events, Enns's (1999) writing draws on excerpts from a communication book written about Tracy by her mother. Even earlier, Alberta-based playwright Heidi Janz wrote a performance called *Return to Sender* with a similar storyline. In a 1997 media interview, Janz explained that the purpose of this play was to "counteract the general mindset of focusing on the disability rather than on the person" (D. Johnston 1997, n.p.). These arts-based (re)worldings of Tracy Latimer's personhood as a lived human experience rather than a prairie-based trope complexify the conditions of possibility for our work in a posthumanist context wherein the politics of location are foundational (Braidotti 2018; Kincheloe 2001); it is possible to think of place as not only where we are, but as an intergenerational network we carry in our bodies through ongoing processes of becoming, which can include enacting different worlds (Simpson 2017). As Lee reminds us (McGregor 2020), art created by and for disabled folks punctures normative ideas of how disability is lived including, in this case, the enactment of mainstream ableism that asserts disability should not be *lived* and ongoingly condones the killing of a rural, disabled girl. Both Enns' and Janz's art insist on reimagining this event in ways that challenge the culturally contingent ableism; their work is a grievous reminder that where "inclusion into the world as it is currently arranged is not possible" for many, including those who have been killed, disability and deaf art delivers new worldly arrangements amid this grief (Chandler 2018, p. 462).

If we are to follow Tuck and McKenzie's (2015) directive to work toward decolonizing research by moving beyond place as a neutral backdrop to our inquiry (p. 18), we must attend to the backdrop of this case; to consider the place of Tracy's murder to be of little or no consequence requires an "erasure of the body" that assumes the body can move, and can *become*, in just any space or place (Cella 2017, p. 285). This assumption is alive today, including through audiocentric assumptions of place neutrality for deaf people. To date, the provincial government has released one set of recommendations concerning deaf education—a 1990 report that asserts that the even distribution of programs and services throughout Saskatchewan is possible (*Preliminary Report on Deaf Education* 1990). Yet, deaf cultural histories in the province tell a different story. These histories include

the 1991 closure of the R. J. D. Williams School for the Deaf which led to an "exodus" of signing deaf leaders, teachers, and professionals from the province (Weber 2021; Weber and Snoddon 2020, p. 602). As a result, the current primary approach to teaching deaf children today is to include them in mainstream systems such as schools, where parents and advocates report being discouraged against using ASL (Saskatchewan Human Rights Commission 2016, pp. 5–7). In reports published in 2016 and 2021, respectively, the Saskatchewan Human Rights Commission acknowledged the isolation of deaf students amid an uneven patchwork of programs for deaf people fueled by conflicting ideologies about the acquisition of speech and language.[5] Concurrently, in recent years, the province's capital city, Regina, has received an inflow of deaf newcomer youth who represent a "fragile beginning of a new generation of deaf community members" (Weber and Snoddon 2020, p. 615). This unique combination of people has been described as a deaf diaspora (p. 615). Some of the folks who make up this diaspora are involved in Weber's Deaf Crows Collective, including through the Collective's flagship performance, *Apple Time*. Described as a "bridge to all realms," *Apple Time* is a production that blends traditional deaf storytelling, puppetry, props, costumes, clowning, and ASL poetry "to create entire new worlds" using no spoken English (Artesian 2018). Such liminal (re)worlding reimagines and reorders the traditional prairie-scape into an evolving setting that includes knowledge and aesthetics from other countries, hybrid translanguaging, and the new, emergent affect and aesthetic that is the result of cultural fractures combined with diasporic assemblage. The works of Enns, Janz, and Deaf Crows Collective remind us that deaf and disabled artists' becomings persistently resist ableist and audiocentric erasures of difference specific to this place in ways that are anything but neutral.

### 3.2. "Ring Those Phones!": Rurality, Charity, and Sanism

The identarian category of "rural" is elusive. Broadly, rurality takes on vastly different shapes and descriptions worldwide, and the contemporary rural–urban dichotomy is contentious (Dymitrow and Brauer 2017, p. 29). To label someone as "rural"—as we do in our labelling of Tracy Latimer as a rural, disabled girl and in our conception of research participants (below) as rural artists—risks imposing a performative identity on someone without their consent (Edensor 2006; Dymitrow and Brauer 2017). Yet, embodiment plays a role in our interaction with space (Cella 2017), and place has a significant impact on our sense of identity (Massey 2005; Stehlik 2017). What is more, disability is constructed differently in different places and spaces (Short et al. 2018). In Saskatchewan, rural disability is constructed through charity.

Since 1976, the Kinsmen Foundation has been hosting a telethon in Saskatchewan called TeleMiracle. As a teenager in the early 2000s, Jones attended a TeleMiracle telethon taping in Regina. On stage was a line-up of volunteers, ready to answer landline telephones. Between performances by (mainly) Canadian talent, hosts would chant "Ring those phones! Ring those phones!"—a jingle familiar to many Saskatchewanians—to urge the live and at-home television audience to call in and donate.[6] Today, the 20-h event continues to follow charity traditions of launching widespread advertising campaigns featuring objectifying images of disabled people. The prominence of whiteness and childhood in these promotional campaigns is a reminder of how infantilization and "inspiration porn" (Young 2014) collude to reassert white dominance, repeating a pattern earlier noted in Tracy Latimer's killing (Peers 2015). Decidedly unpolitical, Tracy's legacy as a person whose access to medical treatment was cut off in childhood through her father's decision to end her life is not acknowledged by TeleMiracle, even though it is plausible that Tracy would have qualified for this charity's funding had she lived. TeleMiracle raises millions of dollars to fund medical treatment and mobility/communication equipment for qualifying Saskatchewanians (TeleMiracle 2022). The telethon's profits fill a glaring social service gaps and represent a necessary route to accessing medical care for disabled adults with few other supports in the province (Loeppky 2022).

The necessity of the telethon for disabled people's survival in the province is also a testament to the dominance of charity tropes in Saskatchewan's cultural milieu. Being charitable is, for many, a necessary condition toward sustaining a rural life (Gibson and Barrett 2018). In their study of small rural missionary hospitals in Canada during the early twentieth century, Vandenberg and Gallagher-Cohoon (2021) report on the legacy of medical missionary work as social welfare in the Canadian Prairies that was foundational to colonial nationhood expansion. The link between medical and social care rooted former premier Tommy Douglas's 1961 introduction of universal health care legislation.[7] Mythology around this achievement is laced with sanism. In a biography of Douglas, writer Margoshes (1999) begins a chapter called "The Battle Over Medicare" with some local folklore:

> ... [Douglas] pays a visit to the mental hospital at Weyburn, where he meets a patient walking along on the grounds.
>
> 'How are you sir, and what is your name?'
>
> 'Oh fine,' the patient said. 'My name is Bob. Who are you?'
>
> 'Oh I'm Tommy Douglas. You know, the premier of Saskatchewan.'
>
> The patient gives him a suspicious glance, then replies: 'That's all right, you'll get over it. I thought I was Napoleon when I came here.' (p. 129)

Douglas reportedly did visit the Souris Valley Extended Care Centre in Weyburn, Saskatchewan (Davies 2019). Prior to its demolishment in 2009, the center was one of the largest institutions in Canada (Leung n.d.). Known for its "psychedelic" legacy of experimental research including LSD treatments (Dyck 2010; Marcotte 2015), the center is a rich starting point for prairie-based mad people's history (Davies 2019; Dyck and Deighton 2017; Leung n.d.; Melville Whyte 2012). In 2002, community members and artists "took over one wing of the derelict [institution]" to stage (The Weyburn Project n.d.), a six-week performance that cast a reflexive view on the hospital as a less-than-charitable institution (Irwin 2019, p. 134). The Knowhere Productions performance began with artists and community members "camping" in one of the site's buildings and rummaging through archival material (abandoned patient records and medical devices) as they contrived a performance in the abandoned building:

> During the two-hour-long performance ... audiences of 25 individuals walked through the once-locked wards, corridors, electric shock treatment rooms, and holding cells. ... Escorted by former psychiatric nurses, they were taken on a journey through 100 years of mental treatment in Saskatchewan. The materiality of the experience (the smells, sounds, textures and light) ... underscored the ... horrible details of medical successes and failed experiments. (Irwin 2019, p. 135)

Such arts-based interventions add context to our bricolage of prairie-based *becomings* and suggest that disability arts movements have a legacy of disrupting and (re)worlding prairie-based colonial ethos, such as that which links charity to care in rural contexts. By offering a new interpretation of prairie-based institutionalization to audiences, the performance centered local experience of difference in an imaginative, affect-evoking way designed to turn our attention toward regional mad histories.

More recently, artists have begun speaking out against the charitable connotations of their lives: Loeppky (2021), a disabled artist-activist, describes today's rendition of this phenomenon as a "crip tax"—extra costs imposed upon disabled Saskatchewanians, who must pay out-of-pocket for much care unless charity projects like TeleMiracle supplement these costs. In contrast to pervasive charitable rhetoric that is so taxing for disabled people, Saskatchewan's first and only disability-led disability arts organization, Listen to Dis (LTD) Community Arts Organization, hosts online cultural events geared toward disabled people's political right to exist and gather. In a 2021 online "Pajama Party Politics" event, LTD members launched a "crip declaration." This ongoing declaration responds to and reimagines disabled Saskatchewanian's place-based experiences. These experiences are described in the organization's annual report as those which position disabled artists as

"crashing headlong into barriers that some claim we've moved past," including but not limited to longstanding charitable tropes and the pervasive crip tax (Listen to Dis 2022, p. 5). The continued legacy of disability arts and culture amid a resoundingly charity-driven ableism demonstrates another mode of (re)worlding well underway on the Canadian Prairies.

## 4. Methodology

Our interdisciplinary artivist research bricolage engages outward-facing intimate scholarship that lends itself to the numerous contexts in which we operate—real, imagined, or otherwise. Our respective disciplines intersect at the nexus of art and activism, offering what Leduc (2016) describes as an "artivist" perspective: a research-laden collision of art, activism, and community as a possible positionality. Specifically, all the research is deeply connected to the prairie-based performers in both the Deaf Crows Collective and Listen to Listen to Dis (2021), the only deaf- and disability-led arts organizations in Saskatchewan. Perhaps it is the action implication of "artivist" that compels us to build something via the process of *becoming-to-know*, which brings us to the multi-method move of bricolage (Kincheloe 2001). Explaining his support for thorough research that spans beyond the limitations of any singular discipline or method, Kincheloe (2001) describes the process ontology (ruptures and insights) one must embrace during the longevity of scholarly work wherein "[w]e occupy a scholarly world with faded disciplinary boundary lines" (p. 683). Rather than bending to methodologies established by a colonial, ableist academe, bricolage responds to interdisciplinary shifts in knowledge production, validation, and rigor. Kincheloe (2001) compares traditional, discipline-bound praxis to worship at a temple: "My argument . . . is that we must operate in the ruins of the temple, in a postapocalyptic social, cultural, psychological, and educational science where certainty and stability have long departed for parts unknown" (p. 681).

We replace Kincheloe's (2001) temple with our own imagined dinner table. Here, too, we might be amid the ruins. Who ever said the table was functional, intact, and inclusive? We can imagine the table as untidy, wobbly, and even overturned; we can allow the table to impress upon us the sinister baggage of "the Prairies" by being a gathering place where there are pieces to pick up, absences to notice, and (re)worlding to observe. We bring our identities, carved as they are in space and time. We layer into these questions about "what we are capable of becoming" amid the impression of "the Prairies" we have gathered thus far (Braidotti 2018, p. 207).

### 4.1. Methods
#### 4.1.1. Interviews

Our research team conducted interviews with 11 rural-identified deaf and/or disabled artists using Zoom and Google Hangouts. Interviewees were recruited in 2021 through the distribution of an email invitation to participate. The invitation included a link to an American Sign Language (ASL) vlog and an English, text-based translation of the invitation. The email was sent by the lead author to 24 deaf- and disability-related organizations across all the provinces and territories. A Qualtrics-based, digital consent form was available to all invitation recipients. The form consisted of ASL vlog sections that were translated into English text to make the consent form accessible in both languages. Through the form participants were given the option to use their real names or pseudonyms. The participants interviewed for this article indicated a preference to use their real first names, and decisions about naming were also confirmed during the interviews. The researchers conducted one-on-one interviews in ASL or English, using a semi-structured interview method. Following Brinkmann (2022), semi-structured interviews are part of a continuum of interview method that fluctuates between structured (with some questions thought through in advance, a designated start and end time, and a decided focus on the participant's story), flexibility (with the researchers' willingness to change course, respond, and be in-relation with participants as the story unfolds), and improvisation (which, Price (2012) argues, is an

essential ingredient to disability-related research that must always be concerned with access). Interviews typically lasted 45 min to one hour. Following the interviews, the research team developed a coding book to conduct a thematic analysis using NVivo. Codes included childhood experiences, community, conceptions of voice, COVID-19, experiences with jam sessions, identity descriptions, rural experiences or experiences of place, and dinner table experiences.

### 4.1.2. Jam Sessions

Sonic improvisation is emerging as a social and artistic practice that resists the "rarefied and exclusive world" (Morris 2022, para. 5) of traditional music composition (Born 2017; Bobier and Ignagni 2021; Lewis 2008; Warren-Crow 2018). The possibilities for interaction, collaboration, and expression in improvisation focus on "not knowing what you will find on the way" (Mattin 2017). A method of sound- and movement-based improvisation that embraces "harsh noise-based research" (Warren and Hopkins 2021) and notes the ableist assumptions around contemporary music composition (Carlson 2016), jam sessions were developed by Pridmore in 2018 for expressing "voice" in different ways. The purpose of jam sessions has evolved in recent years, in tune with emergent incorporations of multisensory experiences in sound-based performances (Morris 2022). Drawing on Alper (2017) and Jones et al.'s (2022) critiques of "giving voice" as well as crip linguistics and disabled languaging (Henner and Robinson 2021), we suggest that jam sessions have drifted away from a concentration on vocality to refer, simply and separately, to expression. Even so, for deaf groups in particular, the conflation of vocality with expression is problematic: in audiocentric cultures, "voice" is a weighty form of cultural capital and a tool used in ongoing attempts to assimilate deaf people into the speaking world, interfering with their vibrant carnivalesque cultural production that does not center on vocal expression (Peters 2009; Weber and Snoddon 2020). For deaf folks, "voice" takes on different meaning than that found in hearing worlds. "Our hands are our voices", Weber (2022) once explained ahead of a jam session. Jam sessions, then, are the act of getting together either online or in person for improvisational expression in a group setting where harmonic, melodic, and rhythmic elements merge to create a whole (Sawyer 2007, p. 2). We create our own unique compositions in many ways, including through embodied actions (e.g., gurgling, clapping, vocal sounds, signing words) and by using technology (e.g., sound-producing software on iPads, mobile phone ringtones, the clanging together of pots and pans). Pridmore (2022) refers to this as "jamming." Between July 2021 and August 2021, Pridmore led four one-hour jam sessions via Zoom with up to 10 participants in each session. Participants in these jam sessions also completed informed consent forms. Jam session participants chose to identify themselves in many ways including by their real first names, by pseudonyms, and by no name (in which case they are known simply as "Participant").

### 4.2. Creative Analytic Practice (CAP)

Creative Analytic Practice (CAP) is a method of knowing via writing in ways that are both creative and analytical and describes our method of experimental writing thus far (Richardson 2000). From the perspectives of four people taking in one another from different sides of the table, from different perspectives, we gather the pieces of our collective knowledge and develop our bricolage through CAP. Foregrounded by feminist poststructuralism's and postmodernism's challenge to rationalism, CAP offers new narratives about the contexts in which writing is produced—from family ties to social movements to disciplinary constraints—that emphasize the critical reflexivity, grounding, and danger of such narratives as "they nest [our] projects ethically" (Richardson 1999, p. 665). Following CAP's imaginative edge, which embraces all forms, including metaphor, we offer both the dinner table and the bricolage as metaphors that uphold this transversal of thought and meet the limits between "what we do" (Greene 2011, p. 3) and the affirmative ethics of "opening outward to the world" (Braidotti 2018, p. 210). CAP invests in multiple, intertextual methods of inquiry and their theoretical underpinnings. Concerned with the

connections between textual data—including textual creation and interpretation of research accounts—bricoleurs become "methodological negotiators" who must step back from traditional ways of knowing and instead make space for the fragmented, layered, and multiple textual representations of something else (Kincheloe 2005, p. 325; Markham 2005). Our "something else" is a constellation of methods, represented through the intimate artivist research we bring to the CAP-anchored imagined dinner table.

## 5. Findings

### 5.1. Jam Sessions as Dinner Table "Flourishing"

During one jam session, participants were invited to imagine a dinner table. The game expanded when they were asked to imagine a potluck to which they brought a dish. A moment later our imagined feast was abundant with layers of sound: popping popcorn, hissing fizzy water, and the slurping licks of eating ice cream. Participants also shared their dinner table experiences. They reported "not being able to keep up with multiple conversations" and experiencing sensory overload. When invited to express these feelings, one participant rocked back and forth, another began hitting their head, and groans filled the air:

> **Participant 1:** [releases a throaty groan]
>
> **Maria: [speaking]** "I'm just frustrated right now"
>
> **Participant 2: [speaking]** "uuuhhhh . . . ." [begins groaning]
>
> **Lee Hope: [speaking]** "inhale, exhale to try and calm down"

In this jam session moment, improvisation was paramount to telling the sound-based story of the dinner table. Each person offered something intimate to the scene, whether through embodied noises or through spoken words. The result is a chorus of agony—a moment of guttural, groaned frustration mixed in with a delicious potluck spread, which lets us in on the mixed feelings some folks carry to the dinner table.

We discovered that it is possible to gain insight into dinner table experience through sound-based improvisation. Despite its absence in high art traditions (Sawyer 2007; Varvarigou 2017), music improvisation is an important method for engaging in collaboration, interdependence, and "flourishing" that involves experimental sound, movements, and meanings (Kuppers 2019, p. 138). We emphasize "flourishing" following Carlson's (2016) connections between musical experience and intellectually disabled people's human flourishing. By this, Carlson refers not to therapeutic musical interventions, but to the ethical significance of letting go of attempts to explain and analyze music and disability in the interest of new modes of solidarity that come with the embodied experience of music making—or, more simply, making sound for the sake of making sound. If part of the problem with mainstream arts is that it emerges from an unspoken agreement between audiences and creators that the output be in some way understandable (Kuppers 2019), we follow in crip arts traditions of usurping understandability in our jam session rendition of dinner table experience. By choosing cacophony over composition, possibilities for new understandings emerge, highlighting the experiences of those who would be excluded from the dinner table.

### 5.2. Dinner Table Experience as Being Overlooked

In research moments where we engaged in word-based inquiry, such as through ASL or English interviews, we were met with narrative dinner table experiences that fluctuate between the metaphoric and the literal. One hearing interviewee, Victor, recalled their experience growing up in the Prairie Provinces in the late 1950s, witnessing a gendered division between those who occupied the metaphorical dinner table in their household:

> The table is an important metaphor in my life . . . for [my mother], the dinner table was always a challenge and very seldom a pleasure, because of course she was expected to do it all and though my dad was helpful, usually at that time we

still had, you know, we had three acres outside of town and my dad had chores. My dad took care of the outside. There is another traditional literary metaphor that goes back to the last century about house and horse, the female perspective is supposedly and this argument uh, embodies by what's inside the house like a dinner table, where a man is outside in the garden, say, or in the barn, or riding a horse or riding to the damsel's rescue or whatever.

Multiple applications of the dinner table as a metaphor allowed for a varied, rich explication of the experiences of deaf and disabled members. By contrast, Mustafa, a deaf participant, explained that many deaf people refer to the literal deferral by loving, hearing members of their families who promise future explanations of family dynamics and tensions, relay of jokes, and explication of arguments, mostly encapsulated in the phrase, "I will tell you later." Only, later never comes as the conversation shifts, its threads forgotten in the effort to keep up. Mustafa elaborates:

> I love my family but its so frustrating . . . so many times I have asked my family member what they are talking about at the table . . . and often they will say 'hold on' and make me wait and wait. . . . But that's what my family does . . . they will not accept the fact that I cannot hear them at all. I have been Deaf since birth . . . my family has asked that I get the cochlear implant and I told them that I can't, I am much too old and it would not work for me.

For Mustafa, the crip affect of missing out is punitive, compounded by familial expectations or normativity that he somehow strive to hear (perhaps through cochlear implants). This example reminds us that dinner table experience is not only the affect of not fitting in because, in part, some bodies do not fit the shape of the imagined dinner table exchange. The impact is the cutting off of deaf family members' access to incidental learning, such as overlapping conversations. This example also reflects research elsewhere that suggests dinner table experience is most prevalent among those from hearing families whose first language is not a signed language (Listman and Kurz 2020).

A hearing interviewee, Kelsey, opened up about the dinner table experience by first describing a meal with other arts-engaged peers who tried to support her eating, but did not think to bring a knife to cut an apple into bite-sized portions that would make the meal more accessible. Later, she expanded on the affect of missed accessibility gestures:

> And also on the other side of things it's like people might, people might be cracking a joke and I might not get it right away because of cognitive processing. So it might take me three days later I'll get that joke but in the moment it's like Oh, I don't know. I don't know what they've said . . . or what it was, right? So that's kind of weird in a sense because you don't know if they are making jokes about you or the people around you.

If mealtime should be an opportunity to both eat and participate in larger group conversations (Meek 2020), the above example illustrates that divisions of labor and expectations of normate comportment related to Western eating rituals pose barriers to equitable participation, even in contexts that mean to be supportive of disability and deaf art. Indeed, disabled and deaf people often "violate" the rules of normate and hearing etiquette (Meek 2020, p. 1690), and our participants' experiences reveal that mealtimes tend to mark a moment where their previously desired, arts-based "disruptions" come to halt, when possible (re)worlding is abruptly grounded by an existent worldly ritual rife with barriers.

More ephemerally, Kelsey expressed concern about not understanding the rhetoric circulating at mealtime until a few days later—a common retrospective element of dinner table experience that comes with the realization of having missed out on conversations. This timelapse reminds us that dinner table experience is a uniquely crip temporal affect of overlooked-ness that lingers after mealtime is over (Meek 2020, p. 1688). As a reviewer of this article pointed out, the grip that dinner table encounters have on experiences may stay with us throughout our lives. This staying power is productive for re-imagining our sense of understandings of events and everyday moments that shape our subjectivities.

Further, the sticky temporal facets of the dinner table experience—its emergence both in-the-moment and as potentially long-held memory—offer new connections to the rubric of crip time.[8] Along with being a "wry reference to the disability-related events that always seem to start late", crip time as affective temporality points to the challenges of normative, linear pacing when we are faced with a combination of ableist barriers and slowness that characterizes the non-normative engagement with an ableist world (Kafer 2013, p. 26). Kafer (2021) suggests we think less about what crip time is and more about what it does (p. 421). To this end: crip time does the work of making space for grief, including grief over lost experiences (Samuels 2017); crip time also encompasses waiting—in Kelsey's case, waiting for meaning to emerge; crip time encompasses the affective, temporal flux of dinner table experience and reminds us that this experience does not end when the table is cleared and the guests depart. We know about the intersections of crip time and dinner table experience because the affective remnants of this phenomenon stretch into other facets of our work, including jam sessions (above) filled with the noises of groans and reminders to breathe.

This cultural disavowal of difference at the dinner table extends to larger arts movements: as LTD artistic director Traci Foster recently pointed out in a public conversation about cripping art and access in Saskatchewan as compared to its urban Canadian counterparts, "We are currently incapable of keeping up, and therefore, of course, we can't catch up" (Listen to Dis 2022). This incapability seems predicated on the major milestones that bring us to the present, including Tracy Latimer's killing and an ongoing telethon that sustains harmful charitable tropes. These important events are exclusive to Saskatchewan and are imprinted on the lives of all prairie people and the backdrop to the collective work disability and deaf artists must do to establish and sustain their presence in this place.[9] Foster's words, then, succinctly characterize the distinctly crip affect of "not keeping up" as something more politically willful than simply being left behind. Disabled and deaf artistry in an overlooked context refuses to operate in parallel to its more highly resourced urban counterparts because it cannot. Instead, prairie-based deaf and disability arts' propulsion happens amidst a uniquely paced, place-contingent resistance and (re)worlding laced with a crip failure (Mitchell et al. 2014) toward "keeping up" or "catching up", and through a deeply experienced enactment of this crip affect, the rural, prairie movement offers something that urban centers cannot.

## 6. Reflection

The findings above offer a glimpse into prairie-based artists' dinner table experience. We note that apart from the golden sheaves and canned harvest vegetables and fruits that made their way to our metaphorical dinner table on some research occasions, the prairie aesthetic is fraught with grave discourses of overlooked-ness that pop up in imagined dinner table settings: a prairie brand of ableism, sexism, racism, and audism that is exacerbated by charity tropes and leaves artists feeling set apart from deaf and disability movements in other places. Through this bricolage of dinner table experience, we reflect on what we and others (deaf and disabled artists) are *becoming* beneath the open prairie sky and amid the highly gendered expectations of rural labor and the charity-riddled community ethos that routinely overlooks their work and results in a sense of being unable to keep up (Braidotti 2018). Places and our orientation to deaf and disabled persons are informed by context; our attempt to understand both disability and deafness as complex processes of becoming is difficult even as the deaf and disability arts sector experiences growth in Saskatchewan (Weber 2013; Weber and Snoddon 2020; Canada Council for the Arts 2021).

Beginning with Tracy Latimer's murder, we map how the world is arranged for deaf and disabled artists in this place, and we assert that the context of artists' work is embedded in challenges about whose life is valuable and therefore grievable (Pullman and Nichols 2015). In other words, the dinner table experience is a useful metaphor in pointing us toward the worthiness emanating from the task of embracing difficulties, collaborating,

and recognizing subjectivity as a collective assemblage. The dinner table can also stand as a representative of the jam session, in that each guest brings an individual "voice" to the jam session's "table", adding to and collaborating with the cacophony in a unique experience of sound making amid crip time, one that is broader than the prescribed limitations of composed music and that allows for communication above organization. From the vantage points of people who have pulled up a metaphorical seat with us and reflected on their lived dinner table experiences, perhaps with the televised cheers and music of TeleMiracle filling the background with noise, we assert that prairie-based deaf and disability art is on a long-time trajectory of becoming that which is rooted in a politics of the overlooked—an affective "flyover" positioning that remains written on the bones of those of us who engage with it, even in varying degrees, and whose history is recalled by the living and memorialized by the dead.

In the spirit of affirmative ethics, overlookedness need not be an affect of mourning or melancholy (Braidotti 2017). For instance, our emphasis on Tracy Latimer through storied fragments is an attempt to nurture an intergenerational connection symbolized by the imagined dinner table as a gathering place that clocks both presence and absence (Braidotti 2018). In part, Braidotti (2017) describes affirmative ethics as a way of being worthy of what happens to us. As bricoleurs we lean into non-linearity and associative thinking and leave our construction of prairie-based deaf and disabled arts decidedly incomplete; we do not consider our partial knowledge to be a blockage (Guyotte and Flint 2021; Thrift 2008). The purpose of experimental bricolage writing, a form of CAP, is to better understand our positioning as people involved—each to different degrees—around the prairie dinner table. This means understanding multiple perspectives as "humble artisan[s]" not as "master thinker[s]" who are part of the dinner event, to borrow Braidotti's (2021) phrase (p. 531). As such, the question of what we want to become is one that spans far beyond our writing—fittingly, given the context of a land whose horizon point can be too far away to spot with human vision. Here, we acknowledge the work of disabled and deaf artists, including Listen To Dis, Deaf Crows Collective, and several disabled artists who remain committed to the political and artistic work of challenging an oppressive prairie ethos and (re)worlding through their uniquely generative positioning as overlooked.

## 7. Conclusions

To close, we return to Ahmed's (2006) question that directs us to the sky and asks "[h]ow we begin to know or feel where we are" (p. 6). We approached this question in a *becoming-to-know* position à la Braidotti (2018) that critically notes the Canadian Prairies' reputation as a pastoral, apolitical backdrop as an ableist, colonial construction. Massie (2010) reminds us of how picturesque biomes on yearly calendars—"an astonishing sunset or skyline framing a golden field of wheat" (p. 172)—offer an "epistemic wallpaper" difficult to peel away from foundational Western imaginaries (Massie 2010; Thrift 2004, p. 585). This is where the significant (re)worlding of deaf and disabled artists takes place. Writing as a bricolage from this place is an experiment in *becoming-to-know* deaf and disability art in the "flyover provinces" over time, through disciplinary combinations (Kincheloe 2005; Yardley 2020). The intimacy of this scholarship is tied up in process ontologies which take place—in this case, beneath the living prairie sky in a place called Saskatchewan—as a starting point for a cumulative bricolage of arts-based knowledge pulled both from our own and others' experiences as we know these so far. We acknowledge that there are several artists and cultural producers doing significant work in prairie-based disability, deaf, and mad arts beyond the scope of this writing. Our bricolage refuses picturesque reductionism, asserting "the frontiers of knowledge work rest in the liminal zones where disciplines collide" and expand complexly, in multiple ways that might transcend disciplinary borders (Kincheloe 2005, p. 689; Thrift 2008). Here, we have offered the metaphor of the dinner table and the memories and temporalities it evokes as representations of that liminal zone to which Kincheloe (2005) refers, where the overlookedness of this place's disability and deaf art ensues an emergent and unique

(re)worlding. Though we know that disabled, deaf, and mad people have always made art, we acknowledge that their contemporary (re)worlding in prairie contexts remains place-based and liminal—often overlooked in, or on the periphery of, the wider world of urban crip artmaking which has gained much attention in Canada in the last decade (Chandler 2019; Chandler et al. 2021; K. Johnston 2012; Orsini and Kelly 2016; Watkin 2022). Ultimately, this nonlinear collection of findings around deaf and disabled art on the Canadian Prairies represents a blend of memory, testimony, and forecasting around the understated political force of deaf and disability arts in an overlooked "flyover" zone.

**Author Contributions:** Conceptualization, C.T.J.; data collection, C.T.J., J.W., A.A. and H.P.; methodology, H.P.; writing—original draft preparation, C.T.J.; writing—review and editing, A.A., J.W. and H.P. Funding acquisition, C.T.J. and H.P. All authors have read and agreed to the published version of the manuscript.

**Funding:** This research was funded through The Social Sciences and Humanities Research Council (SSHRC) Insight Development Grant titled, "Troubling Vocalities: Disability and Deaf Art on the Canadian Prairies" (430-2020-00189). Funding was also provided by The Council for Research in the Social Sciences (CRISS) of the Faculty of Social Sciences at Brock University (October 2020) and by the Social Justice Research Institute (SJRI) through an SJRI Community Engagement Grant (Beyond Niagara) at Brock University (336-242-071).

**Institutional Review Board Statement:** The study was conducted in accordance with the Declaration of Helsinki, and approved by the Institutional Review Board (or Ethics Committee) of Brock University (protocol code 20-261-JONES), approved 21 March 2021.

**Informed Consent Statement:** Informed consent was obtained from all subjects involved in the study.

**Data Availability Statement:** Not applicable.

**Conflicts of Interest:** The authors declare no conflict of interest.

## Notes

1. Commonly lowercase "deaf" is used to refer to audiological impairment while uppercase "Deaf" refers to a cultural group who share beliefs, practices, and a language (American Sign Language). Here, however, we lean into lowercase "deaf" to reflect the recommendation by Friedner and Kusters (2015) to eschew binarization between groups of deaf people and to reflect the notion of deafhood (Ladd 2003) as a state of becoming in which traits belonging to both realities noted by the use of "d" and "D" are often used according to shifting circumstances.

2. The dinner table phenomenon is metaphorical because it represents not only something that happens at dinner time but also the experience of missing out on overlapping conversation that can happen, for example, while a radio is playing during a car ride, children are playing on the playground, and people are gathering during holiday gatherings. Meek (2020) is clear that dinner table experience happens in most any "other instances where a deaf individual interacts with a group of people" (p. 1676), and this experience is especially prominent in the lives of deaf people with hearing families (Listman and Kurz 2020).

3. We acknowledge the work of Monte Hardy for bringing the idea of "dinner table experience" to this project in early 2020 during his work as a research assistant for the The Social Sciences and Humanities Research Council (SSHRC) Insight Development Grant titled, "Troubling Vocalities: Disability and Deaf Art on the Canadian Prairies." At Hardy's suggestion, we orchestrated an ASL-first methodology that was later translated into English.

4. The cultural erasure of Tracy's personhood is also well documented elsewhere (Heavin 2001; Sobsey n.d.). Notably, in a 2003 book chapter describing national reaction to the Latimer case as that which aligns with representational patterns of disability in Canadian literature, Truchan-Tataryn (2003) points to an episode of the Canadian Broadcasting Corporation radio show *Ideas* wherein Hingsburger described a drama class improvisational activity centered on the scene of Tracy's death. The actors initially portrayed Robert Latimer "as a loving father" in a scene about parenthood. When asked to portray Tracy, the actors struggled. "What would she think?" Hingsuberg asked one actor. "Tracy was retarded", the actor replied, "Do they think?" (p. 11).

5. The Saskatchewan Human Rights Commission (2016) report, called *Access and Equality for Deaf, deaf, and Hard of Hearing People: A Report to Stakeholders*, noted that preschool opportunities for deaf children are rare, and services for deaf children entering elementary school are problematic (p. 8). In Saskatchewan Human Rights Commission (2021), an update to the report was published. The update, *Access and Equality for Deaf, deaf, and Hard of Hearing People: Update to Stakeholders 2021*, names actions undertaken to address disparities in programs and services for deaf persons: universal newborn hearing screening (p. 22), implementation of visual bus announcements on City of Saskatoon Transit Service (p. 22), and new provisions around university-

based notetaking support (p. 25) among others. That these actions are brand new offer a glimpse into the audiocentric world deaf folks are navigating in the province.

6    Years later, while teaching an introductory course in critical disability studies, Jones's students were surprised to learn that telethons still existed. United States-based writing on telethons often describes them as events of the past that spectacularized charity-based tropes (Haller 2010) and offered a "'new' freak show" (Smit 2003, p. 689) and left and imprint on our cultural understandings of disability in ways that tend to be oblivious to artistic and cultural movements crafted by disabled folks (Shapiro 1994).

7    Tommy Douglas is widely credited as being the "father of Canada's health-care system" and in 2004 was awarded the honour of "greatest Canadian" by the Canadian Broadcasting Corporation (2004).

8    We wish to thank the editors of this special issue for pointing out that the temporal aspects of the dinner table experience are alive in Kelsey's account. The editors thoughtfully suggested that Kelsey's experience points to new facets of crip time "entangled with audiocentric experiences".

9    In a follow-up email conversation with Foster about this quotation, she explained: "We individually and collectively understand that without the collective presence, our presence is weakened and the fights that we have fought and have made some tiny wins within, will be once again be lost to the community. That is to say, without the onerous work of 'keeping up' with what the funders and other organizations are doing, the shifts we have instigated and insisted on being sustainable will drift back into the ethos of the living skies" (Foster, personal communication, 2 February 2023).

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
