# Peer review of "Dinner Table Experience in the Flyover Provinces: A Bricolage of Rural Deaf and Disabled Artistry in Saskatchewan"

_socsci, doi:10.3390/socsci12030125_

Round 1
Reviewer 1 Report
It has been a great pleasure to review ‘Dinner Table Experience in the Flyover Provinces: A Bricolage of Rural Deaf and Disabled Artistry in Saskatchewan’ for Social Sciences.
This is a beautifully written paper – theoretically engaging, offering a well-crafted and accessible dialogue. I strongly recommend this article for publication. I offer some recommendations for minor revisions below.
The framing of the paper and grounding of the paper in place, with respect to connection of deaf and disabled children’s experiences of being ‘looked over’ with the context of Sask, as a flyover province, was beautifully done – and left quite an impression. The structure and theoretical framing are clear, thoughtful, and unique – I applaud the authors for their craft.
The three markers they discuss – the murder of Tracy Latimer, the continuation of the telethon/role of charities, and their dinner table affect – offer a sense of place and ‘coming to know’ deaf and disability art in Saskatchewan. These offerings speak to each other, each layered by the others, and I invite the authors to integrate them a bit more, referencing each marker (Latimer, telethon, and dinner table) within each discussion. How is the dinner table experience informed by the telethon and Tracy Latimer? How is the telethon shaped by Tracey Latimer’s murder and familial/dinner table experiences? This integration is well done in the closing sections of the article but I wondered if adding in some references within each section might enrich the dialogue a bit more.
Another minor revision I recommend is a bit more elaboration on the history and context in the section on the telethon. The reference to place – home of Tommy Douglas/public health alongside the telethon (representing huge gaps in social services) is important, and I think a bit more detail on the history of health care and role of charities (in relation to social and health care services) as well as implications for individuals and society would be beneficial.
Finally – I invite the authors to reiterate some closing thoughts or reflections on what is possible, made possible – through Deaf and Disabled art in a flyover province. On page 12 – in discussion of Kelsey’s experience, the authors reflect on waiting for meaning to emerge, reminding us that the dinner table experience does not end when guests leave and the table is cleared. This is productive for re-imagining our sense and understanding of events that is seemingly more inline (or in tune) with how we live and experience events and everyday moments; we continue to reflect and re-live many of them throughout our lives. The closing of that paragraph is a quote that states – ‘we are currently incapable of keeping up…we can’t catch up’ by LTD artistic director – and I wonder if the authors could share thoughts on this – while validating the pressure and expectation to ‘keep’ up with other places/people, perhaps they authors could also include recognition that this place, this flyover province, offers something that urban centres are incapable of offering. There is a lovely reference to the expansive potential in the closing paragraph, but as the authors identify, there is something particular to this place being fostered, becoming here – that does not need to be defined or concrete, but I would like to see drawn out a bit. The quote from the article below might be a good place to incorporate this:
“we assert that Prairie-based deaf and disability art is on a long-time trajectory of becoming that which is rooted in a politic of the overlooked—a “flyover” positioning that remains written on the bones of those of us who engage with it, even in varying degrees, and whose history is recalled by the living and memorialized by the dead.” *what does this offer and/or make possible – that is tied to place (and social relations in place)?
The above are offerings, but the article as it stands is very strong and makes an excellent contribution to many fields of study; arts-based research, disability studies, social justice research, and human geography – to name a few. I will definitely be recommending this article to colleagues and students, and will be incorporating it into my methodologies course once it is available.
Author Response
Thank you very much for your review.

Reviewer 2 Report
A very well written article with a strong theortical framework supporting this important research topic. A lot of work have gone into this paper and I commend the authors for addressing this topic. A couple of suggestions for improvement are listed below:
Literature Section
I did not see hypothesis or clear research questions related to this project.
Methods
It doesn’t mention if this research project was approved by the Human Subjects office.
Needs a clearer description of the method for possible replication. No mention of what kind of research (qualitative, mixed, etc). No demographic information of the participants? Were interviews one-on-one or through focus group? Whose work (methods) are you using? How was the invitation conducted? Mail, word of mouth or email?
I would suggest moving the Jam Session and CAP information to the literature section at the end just right before the Methods section then describe how you used Jam Session and CAP as part of your methodology in the methods section.
Findings
Move the description of jam sessions to the Methods section and focus only on the results in the “findings” section.
Names were used in the findings. It is unclear if they are actual name or pseudonym names? If so, you need to explain that in the Methods section.
I am not sure what the journal formatting guideline is related to quotes from participants but it’s hard to follow when it is a quote. In some journals, quotes are italicized or indented on both sides.
Author Response
Thank you very much for your review.
